# mTOR Activity and Autophagy in Senescent Cells, a Complex Partnership

**DOI:** 10.3390/ijms22158149

**Published:** 2021-07-29

**Authors:** Angel Cayo, Raúl Segovia, Whitney Venturini, Rodrigo Moore-Carrasco, Claudio Valenzuela, Nelson Brown

**Affiliations:** 1Center for Medical Research, University of Talca School of Medicine, Talca 346000, Chile; acayo@utalca.cl (A.C.); ralemilio@gmail.com (R.S.); whitneyventurini@gmail.com (W.V.); cvalenzuela@utalca.cl (C.V.); 2Department of Clinical Biochemistry and Immunohematology, Faculty of Health Sciences, University of Talca, Talca 346000, Chile; rmoore@utalca.cl

**Keywords:** mTOR, autophagy, senescence

## Abstract

Cellular senescence is a form of proliferative arrest triggered in response to a wide variety of stimuli and characterized by unique changes in cell morphology and function. Although unable to divide, senescent cells remain metabolically active and acquire the ability to produce and secrete bioactive molecules, some of which have recognized pro-inflammatory and/or pro-tumorigenic actions. As expected, this “senescence-associated secretory phenotype (SASP)” accounts for most of the non-cell-autonomous effects of senescent cells, which can be beneficial or detrimental for tissue homeostasis, depending on the context. It is now evident that many features linked to cellular senescence, including the SASP, reflect complex changes in the activities of mTOR and other metabolic pathways. Indeed, the available evidence indicates that mTOR-dependent signaling is required for the maintenance or implementation of different aspects of cellular senescence. Thus, depending on the cell type and biological context, inhibiting mTOR in cells undergoing senescence can reverse senescence, induce quiescence or cell death, or exacerbate some features of senescent cells while inhibiting others. Interestingly, autophagy—a highly regulated catabolic process—is also commonly upregulated in senescent cells. As mTOR activation leads to repression of autophagy in non-senescent cells (mTOR as an upstream regulator of autophagy), the upregulation of autophagy observed in senescent cells must take place in an mTOR-independent manner. Notably, there is evidence that autophagy provides free amino acids that feed the mTOR complex 1 (mTORC1), which in turn is required to initiate the synthesis of SASP components. Therefore, mTOR activation can follow the induction of autophagy in senescent cells (mTOR as a downstream effector of autophagy). These functional connections suggest the existence of autophagy regulatory pathways in senescent cells that differ from those activated in non-senescence contexts. We envision that untangling these functional connections will be key for the generation of combinatorial anti-cancer therapies involving pro-senescence drugs, mTOR inhibitors, and/or autophagy inhibitors.

## 1. Introduction

It has now become evident that cellular senescence is not a homogeneous phenotype. Rather, several types of senescence, with different consequences for tissue homeostasis, can be recognized [1,2,3,4,5,6]. Features such as the nature of the senescence-inducing stimulus, the cell of origin, and the ability of senescent cells to synthesize and release bioactive molecules have all been described as determinants of the ultimate effects of senescent cells in complex tissues [4,7,8,9]. Underlying the diversity of senescent cells are complex cell-autonomous synthetic and catabolic pathways. Understanding these metabolic changes is particularly relevant for drug-induced senescence, which is often encountered in the context of cancer treatment [10,11,12,13,14]. By modifying the metabolic status of senescent cells, one would expect to tilt the balance towards less harmful and more therapeutically effective forms of senescence. In this review, we begin with a general account of cellular senescence, in order to then discuss the role of mTOR and autophagy in this process. In light of the extensive number of cellular systems in which cellular senescence has been studied, therapies that induce senescence as part of their anti-cancer effects, such as cyclin-dependent kinase 4 and 6 (CDK4/6) inhibitors [15,16,17,18,19,20], will be used to illustrate key concepts and functional connections.

## 2. Cellular Senescence in a Nutshell

Cellular senescence is a unique form of cell cycle arrest triggered in response to a wide range of physiological and pathological stimuli [5,9]. The initial description of cellular senescence was based on analyses of primary human fibroblasts propagated in culture [21,22]. After undergoing several rounds of division, these cells enter a state of proliferative arrest characterized by an inability to respond to growth factors [2]. The finite number of doublings before exiting the cell cycle, known as the Hayflick limit, was later shown to be the result of cell division-driven telomere shortening [3]. Importantly, subsequent studies demonstrated that a similar phenotype can be triggered in response to the stress that accompanies oncogene activation, DNA damage, reactive oxygen species (ROS) generation, mitochondrial dysfunction, chemotherapeutic drugs, and other stimuli [23]. To distinguish them from the process of “replicative senescence” that occurs in primary human cells, these various forms of senescence were grouped under the name of “premature senescence” or “stress-induced senescence” [1]. What most of these forms of cellular senescence have in common is the activation of the DNA damage response [24,25,26], although some stimuli, including the exposure to some chemotherapeutic drugs, induce senescence with no evidence of direct genomic damage [27].

As the exit from the cell cycle in senescent cells is commonly established and maintained through the activation of tumor suppressor pathways centered on p53 and/or pRb [28], it was rapidly recognized that cellular senescence represented a cell-intrinsic mechanism that limits the proliferation of damaged and potentially cancerous cells [4]. Thus, much like apoptosis, cellular senescence acts as a barrier that cells must overcome in order to become immortalized and transformed, explaining the high frequency at which these pathways become disrupted in human cancers [29,30].

Unlike cells undergoing other forms of cell cycle arrest, senescent cells acquire a set of unique morphological and functional features [31]. Under the microscope, they appear as enlarged and flattened cells, with large nuclei and multi-vacuolated cytoplasms [32]. A higher number of lysosomes, which results in higher β-galactosidase activities at sub-optimal pH [33,34], is also typical of most senescent cells. Another feature of senescent cells is the accumulation of senescence-associated heterochromatin foci (SAHF), regions of chromatin condensation associated with the silencing of several genes involved in proliferation [35,36]. It is important to stress, however, that these microscopic features, along with the activation of the p53 and pRB pathways, are not necessarily specific to cellular senescence. Therefore, several markers of senescence are commonly required to confirm the presence of senescent cells [5].

A turning point in the field was the realization that cellular senescence also occurs in the context of complex tissues [37,38,39,40]. It is now accepted that senescent cells participate in processes as diverse as embryonic development, wound healing, and tissue repair [4,8]. Senescent cells also accumulate in aging tissues, contributing to the tissue and organ dysfunction that accompanies organismal aging [7,9]. Indeed, persistent senescent cells have been linked to chronic inflammation and a decrease in lifespan [37]. Accordingly, targeting senescent cells in aged mice, using genetic or pharmacological approaches, attenuates chronic inflammation and restores fitness, demonstrating a causal link between the accumulation of senescent cells and age-related decay [14]. Therefore, targeting senescence could represent a powerful strategy to prolong the period of time free of chronic diseases [41].

In line with these in vivo effects, and far from being passive bystanders, senescent cells can synthesize a wide variety of bioactive molecules that can then be secreted to the extracellular environment [13]. Secreted factors include growth factors, cytokines, chemokines, proteases, and components of the extracellular matrix [11]. As expected, this “senescence-associated secretory phenotype (SASP)” greatly expands the effects of senescent cells arising in complex tissues [12,42,43]: while some SASP factors perpetuate, or even promote, senescence [44], others can stimulate proliferation or enhance the migration of non-senescent neighbor cells [10]. Of note, due to the trophic and pro-inflammatory actions of many SASP components [11], senescent cells arising in tissues can promote an inflammatory microenvironment that, in the long run, could facilitate tumorigenesis [45,46]. Similarly, senescent cells arising in the tumor microenvironment, after exposing cancer cells to senescence-inducing chemotherapeutic drugs or radiation, could perpetuate a post-therapy inflammatory state leading to cancer recurrence [47]. Thus, in contrast to the view of cellular senescence as a cell-autonomous mechanism that suppresses the proliferation of cells at risk of malignant transformation [33,48], the long-term persistence of senescent cells in tissues may have non-cell-autonomous detrimental effects [49], likely reflecting the chronic actions of SASP profiles enriched in inflammatory molecules [50,51]. It is important to mention, however, that senescent cells do have the ability to signal their own removal through the secretion of factors that attract and activate immune cells [52]. This has led to the notion that chronic senescence, with its detrimental consequences for tissue homeostasis, may reflect, at least in part, a decline in the ability of the immune system to remove senescent cells from tissues [49,53].

Underlying senescence phenotypes, including the SASP, are complex changes in cellular metabolism [54,55]. In fact, dysregulation of cellular metabolism is a hallmark of cellular senescence [52], a feature partially attributed to epigenetic modifications [56]. For example, studies carried out in models of oncogene-induced senescence (OIS) have revealed the existence of a predominantly mitochondrial oxidative metabolism with increased oxygen consumption, ATP production, and lipid catabolism, in senescent cells [57,58,59]. In other models, however, senescent cells seem to acquire a more glycolytic state, even in the presence of high oxygen levels [60]. Among the signaling circuitries involved in the modulation of catabolic and synthetic processes, as well as the maintenance of the energy balance, those centered on mTOR have slowly emerged as key determinants of senescent phenotypes [61].

## 3. mTORC1 Activity in Senescent Cells

mTOR (mechanistic target of rapamycin) is a serine/threonine kinase involved in the integration of multiple metabolic and growth-promoting signals [62]. Upon activation, mTOR promotes cell growth and survival through the regulation of protein synthesis and other biosynthetic processes while limiting autophagy-mediated catabolism [63,64]. An extensive biochemical characterization has revealed the existence of two mTOR-containing multiprotein complexes (mTORC1 and mTORC2), each distinguished by its own set of accessory proteins and differential sensitivity to the drug rapamycin, and each involved in the regulation of unique aspects of cell growth [65]. On the one hand, mTORC1 (containing at its core mTOR, mLST8, FKBP12, DEPTOR, and the scaffold protein RAPTOR) integrates nutrient, growth-promoting, and stress-related signals and translates these inputs into adaptive responses that tune the balance between anabolism and catabolism [66,67]. mTORC2 (containing mTOR, mSIN1, DEPTOR, and the scaffold protein RICTOR), on the other hand, orchestrates dynamic rearrangements of the cytoskeleton and the activation of pro-survival pathways in response to growth-promoting signals [68,69]. Unlike mTORC1, which can be acutely inhibited by rapamycin, mTORC2 is inhibited only upon chronic exposure to this drug [69,70,71]. Interestingly, mTOR has been found in several subcellular compartments, including mitochondria, the plasma membrane, the endoplasmic reticulum, the nucleus, and lysosomes [72,73,74]. While the functions of mTOR at different subcellular localizations remain incompletely understood [72], mitochondria-localized mTOR seems to promote the synthesis of the mitochondrial transcription factor A (TFAM), mitochondrial ribosomal proteins, and components of complexes I and V, necessary for the coordination between mRNA translation and energy production [75,76,77]. Similarly, lysosomal mTOR seems to be essential for autophagy and lysosomal biogenesis [78].

Activation of mTORC1 by nutrients and/or growth factors typically promotes protein synthesis while, at the same time, suppressing catabolic processes [79,80]. For example, mTORC1 promotes the de novo synthesis of lipids through the activation of the sterol regulatory element-binding protein (SREBP), an important transcriptional regulator of lipogenic genes [81,82]. Additionally, the S6K kinase, a downstream target of mTORC1, facilitates the de novo synthesis of nucleotides through phosphorylation of carbamoyl-phosphate synthetase 2, aspartate transcarbamylase, and dihydroorotase enzymes (CAD), as well as the transcriptional activation of pentose phosphate pathway (PPP) enzymes that produce ribose and pyrimidines [83,84]. Moreover, protein synthesis is enhanced by S6K and the mTOR-mediated phosphorylation and inhibition of 4E-BP, events required for the initiation of cap-dependent translation and ribosome biogenesis [85,86,87]. These and other mTORC1-dependent processes provide the necessary building blocks to sustain cellular growth and proliferation [88,89]. It is therefore hardly surprising that mTOR activity is upregulated by oncogenic signals in the majority of human malignancies [90].

Accumulating evidence indicates that mTORC1 activity is required for the orchestration of cellular senescence [91]. For example, senescence can be delayed, and some of its features even reversed, following inhibition of mTORC1 in primary fibroblasts undergoing oncogene-induced senescence [92,93]. A similar senescence-impairing effect was documented in primary fibroblasts that were pre-treated with rapamycin before entering replicative senescence [94], although inhibition of mTORC1 resulted in a state similar to quiescence [95] and therefore did not confer a proliferative advantage to cells [96,97]. Some of these findings have been replicated in models of senescence caused by ectopic expression of cyclin-dependent kinase (CDK) inhibitors (such as p21 and p16), whereby the pharmacological inhibition of mTORC1 led to a delay, or even a suppression, of the senescent phenotype [96,98,99]. Interestingly, it has been proposed that cell cycle arrest coupled to hyperactive mTOR leads to cellular senescence, whereas cell cycle arrest accompanied by low levels of mTOR activity leads to quiescence [100,101]. For example, cell cycle arrest secondary to stabilization of p53 led to inhibition of mTOR and quiescence in WI-38 cells. On the other hand, doxorubicin-mediated cell cycle arrest, which was not accompanied by mTOR inhibition, led to senescence instead of quiescence [102].

Recently, it was reported that rapamycin-mediated inhibition of mTOR could abrogate senescence induced by doxorubicin or hydrogen peroxide in mesenchymal stem cells purified from human umbilical cords, an effect attributed to a reduction in the levels of DNA damage [103]. It has also been revealed that the delayed replicative senescence in endothelial cells caused by inhibition of mTORC1 could be modulated by microRNAs through their influence on PTEN [104,105]. Finally, evidence has emerged indicating that mTOR may be involved in the maintenance of senescence in models of cardiomyocyte differentiation [106,107,108]. Thus, exposure of human cardiac progenitor cells to doxorubicin led to ROS accumulation and DDR activation, promoting cellular senescence [109,110]. In addition, accumulation of ROS in cardiomyoblasts and primary cardiomyocytes induces a DNA damage response and typical senescence characteristics. In this setting, inhibition of mTOR prevents mitochondrial dysfunction and induction of senescence [111]. These data are in agreement with the fact that treatment of senescent cells with rapamycin, a known activator of autophagy, can reduce mitochondrial mass and ROS generation, as well as several other markers of cellular senescence [112,113].

Interestingly, mTORC1 inhibition can also suppress the expression and secretion of inflammatory cytokines in senescent cells by selectively blocking the translation of membrane-bound IL-1 alpha and by reducing the transcriptional activity of NF-kappa B [114], which in turn leads to a reduction in the expression and secretion of IL-6 and IL-8 [115]. The effects of mTORC1 inhibition on the secretory phenotype might reflect, at least in part, the ability of 4EBP1, one of the substrates of mTORC1, to regulate the phosphorylation of the RNA-binding protein ZFP36L1 during senescence, thus inhibiting its ability to degrade the transcripts of SASP components [116].

Overall, the dependence of senescent cells on mTOR activity could have important consequences for therapies involving pro-senescence drugs. Notably, several studies suggest that CDK4/6 inhibition activates mTORC1, which in turn may be necessary for the survival, and therefore persistence, of senescent cells [117]. Accordingly, the combination of mTORC1 and CDK4/6 inhibitors reduces cancer cell growth more effectively and delays resistance to therapy in models of breast carcinoma, anaplastic thyroid carcinoma, and cholangiocarcinoma [118,119,120]. Other studies have pointed to similar synergic therapeutic activities between mTORC1 and CDK4/6 inhibitors, particularly in the context of breast tumors [121,122,123]. Interestingly, the resistance to CDK4/6 inhibition commonly observed in pancreatic ductal adenocarcinoma (PDAC) was attenuated by mTOR inhibitors [124]. Indeed, the combination of mTORC1 inhibitors and CDK4/6 inhibitors had a potent activity across a large number of patient-derived models of PDAC and breast cancer [118,125]. Similarly, PI3K inhibitors, which are expected to reduce mTORC1 activity, acted synergistically with CDK4/6 inhibitors in models of mesothelioma and *K-Ras*-mutated non-small cell lung cancer (NSCLC) [126,127]. In contrast to these reports, which highlight a positive role of mTOR in senescence induced by CDK4/6 inhibition, a significant inhibition of mTORC1 signaling was observed in melanoma and glioma cells after treatment with the CDK4/6 inhibitor palbociclib [16,18,128].

Taken together, most reports suggest a positive role of mTOR in the implementation of different aspects of cellular senescence, although the extent to which mTORC1 activity is required varies depending on the modality of senescence and/or the cell lineage involved. Thus, the effects mTOR inhibition can range from a reversion of senescence features, as occurs in models of OIS [92], to reduced viability of senescent cells, as occurs in models of senescence induced by CDK4/6 inhibition, or more subtle changes in SASP profiles [3,129]. Figure 1 depicts some of the upstream inputs and downstream effectors of mTORC1 in senescent cells; the functional connection between mTORC1 and autophagy is also shown (see below).

## 4. Autophagy in Senescent Cells

Autophagy is a highly regulated catabolic process in which cellular components are targeted for lysosome-mediated degradation [130,131]. Basal levels of autophagy serve as a quality control mechanism that prevents the accumulation of protein aggregates and damaged organelles [132]. In metabolically challenged or damaged cells, on the other hand, an increased autophagic flux provides basic metabolic substrates necessary for short-term survival [133]. Under these circumstances, autophagy is thought to maintain the mitochondrial function, the energy balance, and lipid metabolism and also provides substrates that feed metabolic pathways needed for amino acid and nucleotide synthesis [134,135,136,137]. At the organismal level, autophagy constitutes an important mechanism of adaptation to starvation and metabolic disturbances; it also serves as a mechanism that prevents the emergence of neurodegenerative diseases driven by the reduced elimination of misfolded or aggregated proteins [138,139]. Notably, the role of autophagy in cancer appears to be context-dependent [140,141]. While autophagy deficiency may promote tumorigenesis in some models [142,143,144], fully transformed cancer cells still depend on autophagy to withstand the metabolic stress present in tumor microenvironments [145,146].

A hallmark of the autophagic process is the enclosure of cellular components by double-membrane structures known as autophagosomes [147]. The formation of autophagosomes involves an orchestrated series of events, including the initial nucleation of an isolating membrane, or phagophore, followed by elongation and membrane closure [148]. Subsequently, autophagosomes are fused with lysosomes [149,150], forming secondary vacuoles known as autolysosomes, in which the contents and membranes of autophagosomes are degraded [151]. This dynamic process of degradation is carried out by the sequential involvement of several functional complexes formed by proteins encoded by evolutionarily conserved genes known as autophagy-related genes (ATG) [152].

As it has already been mentioned, basal and induced modalities of autophagy can be distinguished depending on nutritional conditions [153]. Under nutrient-rich conditions, autophagy is greatly suppressed, although it still occurs at constitutively low levels [154]. In cells growing under starving conditions, on the other hand, autophagy rates are induced in order to maintain the energy balance and a substrate reserve within cells [155]. Not surprisingly, mTOR-centered signaling is also involved in the regulation of autophagic flux [156]. Thus, the inhibition of mTORC1 that follows nutrient or growth factor deprivation is a key event associated with higher rates of autophagy [157,158]. Mechanisms involved in autophagy upregulation in the context of mTORC1 inhibition include the activation of proteins necessary for autophagosome formation [159,160], as well as the activation of transcription factors required for lysosomal biogenesis [161,162]. For example, inhibition of mTORC1 fails to produce the phosphorylation-dependent inactivation of ULK1 and ATG13, two early effectors in autophagy induction that, together with FIP200 and ATG101, drive autophagosome formation [163,164,165]. Similarly, mTORC1 inhibition driven by nutrient deprivation leads to nuclear translocation of the transcription factors TFEB and TFE3, with the subsequent activation of genes required for lysosomal biogenesis [162,166,167]. An increased lysosomal mass results in increased rates of protein degradation and the reconstitution of a pool of amino acids that allows the reactivation of the mTORC1 pathway after prolonged starvation [78,168,169,170].

It has now become evident that lysosome-located mTORC1 constitutes a key molecular node that connects autophagy and the amino acid availability that is required for the survival of cells growing under metabolically challenging conditions [73,171,172,173]. Under metabolically favorable conditions, mTORC1 is recruited to the lysosomal membrane by Rheb [174], which binds to GTP and activates the mTORC1 complex [175]. Upon growth factor or nutrient deprivation, on the other hand, the GTPase-activating complex, TSC1/TSC2, converts active lysosomal Rheb-GTP into inactive Rheb-GDP, leading to mTORC1 inactivation and autophagy upregulation [176]. Anchoring of mTORC1 to the lysosomal membrane is mediated by the actions of Rag-GTPases and the pentameric Ragulator complex [177,178,179]. Indeed, heterodimeric Rag-GTPases are essential elements of the nutrient sensing machinery [180]. By undergoing changes in activity, driven by the amino acid and the nutritional status, Rag-GTPases modulate the recruitment of mTORC1 to the lysosome membrane, enabling its activation by lysosomal Rheb [64,177] (Figure 1).

Although autophagy was initially thought to be a process that counteracted cellular senescence by removing damaged macromolecules or organelles, research from Narita’s lab indicated that autophagy upregulation was required for the implementation of oncogene-induced senescence (OIS) [181,182,183]. Thus, inhibition of autophagy in primary human fibroblasts delayed the onset of senescence driven by the overexpression of oncogenic *H-Ras* [93]. As a result, autophagy-deficient, *H-Ras*-expressing fibroblasts were rendered more proliferative than autophagy-proficient controls [93]. Similarly, studies carried out in adult human lung fibroblasts (HLF) showed that autophagy upregulation was associated with senescence induction in models of myofibroblast differentiation, fibrosis, and mesenchymal stem cells [184,185]. Interestingly, autophagy in the context of OIS was later shown to generate a high flux of recycled amino acids and other metabolites, which are subsequently used by lysosome-bound mTORC1 for the synthesis of SASP factors such as the cytokines IL-6 and IL-8 [186,187]. Indeed, a TOR-autophagy spatial coupling compartment (TASCC) responsible for the synthesis of some SASP factors was proposed [186]. Thus, a catabolic process (autophagy) can be coupled to an anabolic process (mTORC1-dependent protein synthesis) in order to effectively coordinate the production of SASP proteins [181,187,188]. Attractive as this model may seem, it is presently unclear whether it can be extrapolated to other modalities of senescence, particularly models of drug-induced senescence in the context of cancer treatment [189,190,191]. In addition, the model must be reconciled with the fact that autophagy upregulation is commonly accompanied by mTORC1 inhibition in non-senescent cells [192,193,194]. As it has already been mentioned, mTORC1 inhibition secondary to a reduced availability of metabolic substrates and/or growth factors is accompanied by high rates of autophagy [159,160]. Therefore, while mTORC1 is an upstream regulator of autophagy in most cells, at least in certain types of senescent cells, autophagy can feed the mTORC1 complex, a step necessary to implement the SASP [184,195].

In contrast to autophagy-dependent senescence, models of drug-induced senescence suggest that autophagy is not involved in the implementation of senescence phenotypes per se but constitutes, instead, a maintenance or survival mechanism [181,196]. For instance, although induction of senescence in mammary epithelial cells exposed to pro-senescence CDK4/6 inhibitors, such as palbociclib, is accompanied by high rates of autophagy, the concomitant inhibition of CDK4/6 and autophagy does not render these cells more proliferative but exacerbates the senescent phenotype [197,198] or leads to cell death [199]. In line with these observations, autophagy inhibition significantly improves the efficacy of CDK4/6 inhibitors against breast cancer [200]. Interestingly, there is also evidence suggesting that autophagy induction can protect cancer cells from chemotherapy-induced apoptosis by promoting senescence [183]. Overall, these studies suggest that autophagy upregulation in most senescent cells, including senescent cancer cells, represents a metabolic adaptation that promotes cell viability [201,202], highlighting autophagy as a promising target in cancer cells undergoing drug-induced senescence [203,204,205,206]. Nevertheless, a note of caution must be added here: while apoptosis may be induced in senescent cells exposed to autophagy inhibitors [207], an exacerbation of senescence, with its detrimental consequences for tissue homeostasis, may also occur [181,208]. Between apoptosis and an exacerbation of senescence, the blockade of autophagy could also have more subtle consequences, such as SASP alterations that may be beneficial or detrimental depending on the cancer type and context. Differences in the outcome of autophagy inhibition likely reflect differences in the genetic background of cells undergoing senescence, as well as differences in the specific senescence-inducing stimulus (Figure 2).

## 5. Conclusions and Future Directions

Several lines of evidence suggest that autophagy upregulation observed in senescent cells [196] is not accompanied by a decrease in mTORC1 activity (as it does occur with the upregulation of autophagy observed in non-senescent cells under conditions of nutrient deprivation). Until now, however, it remains unknown whether these changes are universal to all models of cellular senescence. For instance, it is possible that the mTORC1 functional status in senescence, and its relationship with autophagy, may differ depending on the type of senescence (replicative senescence, oncogene-induced senescence, drug-induced senescence) and the cell type under study (transformed, immortalized, or primary cells) (Figure 1). Moreover, it is not known whether inhibiting autophagy, mTORC1, or both can lead to consistent alterations in the phenotypic features of senescent cells, such as the SASP. This is important, taking into consideration that mTORC1 inhibition, as well as autophagy inhibition, has been considered a therapeutic alternative in cancer. Untangling these functional connections is therefore imperative for the generation of combinatorial anti-cancer therapies involving pro-senescence drugs, mTORC1 inhibitors, and/or autophagy inhibitors.

## Figures and Tables

**Figure 1 ijms-22-08149-f001:**
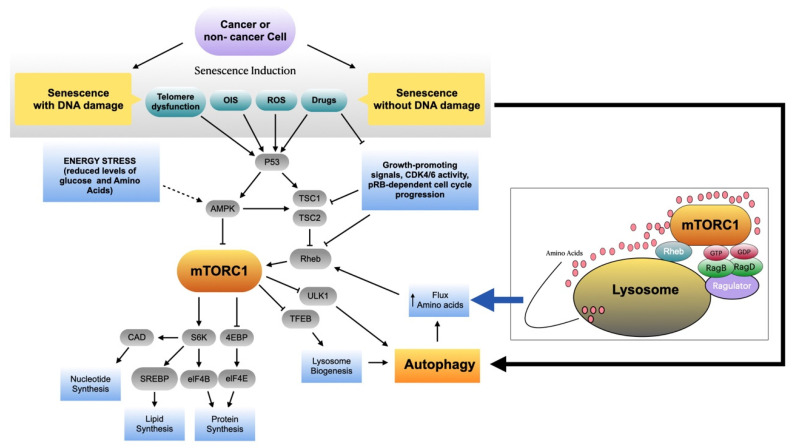
mTORC1 and autophagy as modulators of cellular senescence. Cancer and non-cancer cells can be rendered senescent by different stimuli. Some of these stimuli induce senescence by activating the DNA damage response, while others, including some chemotherapeutic drugs (e.g., CDK4/6 inhibitors), do not seem to require this step to activate the senescence program. It has been shown that p53-dependent responses induce the inhibition of mTORC1 through pathways that impinge on the tuberous sclerosis complex (TSC1/TSC2) or, indirectly, through activation of AMP-activated protein kinase (AMPK). Similarly, pro-senescence stimuli coursing with absence of DNA damage (e.g., senescence induced by CDK4/6 inhibitors) would be expected to lead to mTORC1 inhibition by relieving Rheb GTPase. As mTORC1 activity is elevated in senescent cells, it is thought that inputs for mTORC1 activation are provided by senescence-associated autophagy. By undergoing activation driven by the amino acids released by autophagy-mediated recycling, Rag-GTPases orchestrate the recruitment of mTORC1 to the lysosome membrane, a step that enables the activation of mTORC1 by the lysosomal GTPase Rheb. Known substrates of mTORC1 complexes include the protein S6 kinase (S6K) and the eukaryotic translation initiation factor 4E binding protein (4E-BP). Phosphorylation of these proteins by mTORC1 leads nucleotide synthesis by CAD (carbamoyl-phosphate synthetase 2, aspartate transcarbamylase, and dihydroorotase), lipid synthesis by sterol regulatory element binding protein (SREBP), and protein synthesis by the combined actions of S6K, eIF-4B, and eIF-4E. Notice that, in non-senescent cells, active mTORC1 complexes also inhibit autophagy by blocking the activity of proteins responsible for the initiation of the process (e.g., ULK1) and by sequestering transcription factors needed for lysosomal biogenesis (TFEB).

**Figure 2 ijms-22-08149-f002:**
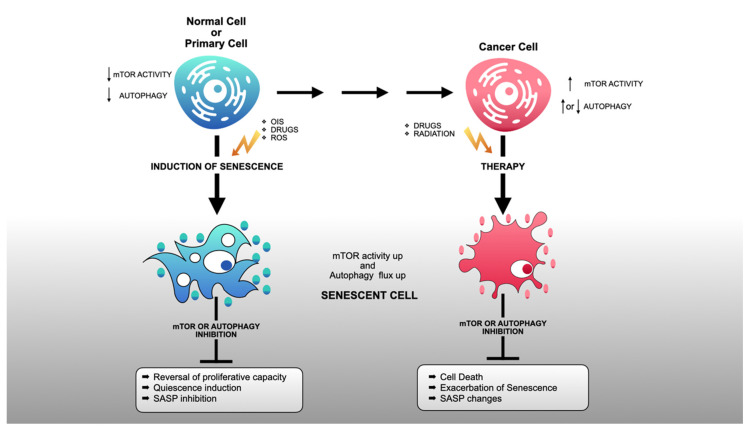
The effects of inhibiting autophagy or mTORC1 in primary or cancer cells undergoing senescence. Senescent cells subjected to mTORC1 or autophagy inhibition can have different fates depending on the senescence-inducing stimulus and tumorigenic status. As illustrated at the top of the figure, normal or primary cells contain generally reduced, although variable, basal levels of mTORC1 activity and autophagy. In contrast, most fully transformed—and genomically unstable—cancer cells are characterized by high levels of mTORC1 activity and variable levels of autophagy. The dynamic variation in the levels of autophagy in cancer cells likely reflects the effects of oncogenic stress and the metabolic challenges encountered at the tumor microenvironment. As shown in the middle of the figure, induction of senescence in normal (diploid) primary cells is generally accompanied by an upregulation of both mTORC1 activity and autophagy. Autophagy in this setting may be functionally coupled to mTORC1-dependent synthesis of SASP components, a functional interaction that is required for the implementation of the senescence program. As a consequence, inhibition of mTORC1 or autophagy can block senescence altogether or specifically abrogate the synthesis of SASP components. Unlike primary cells, cancer cells rendered senescent by pro-senescence chemotherapeutic drugs upregulate autophagy while retaining high levels of mTORC1 activity. In this setting, mTORC1 and autophagic activities may be functionally uncoupled, with each activity contributing independently to the maintenance or survival of senescent cells. As a consequence, inhibition of mTORC1 or autophagy would be expected to reduce the survival of senescent cells, exacerbate the senescence phenotype, or lead to more subtle changes in SASP profiles, depending on the biological context. Small arrows pointing up mean increased activity and small arrows pointing down mean decreased activity.

## Data Availability

Data sharing is not applicable.

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
