# Peer review of "mTOR Activity and Autophagy in Senescent Cells, a Complex Partnership"

_ijms, 2021, doi:10.3390/ijms22158149_

Round 1

Reviewer 1 Report

The manuscript by Cayo et al describes a complex relationship between senescence, mTOR activity and autophagy in cancer and normal cells. In general, this work is well-written, well-organized and provides a comprehensive overview of mTOR activity and autophagy in senescent cells. I would only suggest to provide a figure that describes a molecular pathway of cellular senescence with modulating activity of mTOR.

Author Response

We thank reviewer #1 for his/her encouraging comments. As suggested, we have now included a new figure (Figure 1 in this new version of the manuscript) depicting a possible mechanism that may explain the increased levels ofmTORC1 activity observed in senescent cells. In this model, inputs for mTORC1 activation are provided by senescence-associated autophagy. Notice that, the activation of classical senescence-related pathways –such as those centered on p53 or pRB– and their known downstream effectors, cannot, by themselves, explain the increases in the levels of mTORC1. This explains the need to incorporate the autophagy-mTORC1 axis in this figure.

Reviewer 2 Report

This review summarizes the reports of literature on mTORC1 activity and autophagy in several forms of senescence and examined their functional connections to suggest autophagy regulatory pathways that are different in senescent and normal cells. The authors also tried to check the senescence-specific regulatory link between mTORC1 activity and autophagy in cancer cell context in an attempt to suggest alternative cancer therapeutics. In general, the manuscript is written well and reasonably organized. However, the literature search appears to be carried out rather poorly and therefore, this review provides limited information and poor insight in the roles of mTORC1 and autophagy and their relationships at molecular level. Furthermore, the authors repeatedly mentioned the possible difference in the relationship between mTORC1 and autophagy among different types of senescence, but failed in suggesting any clue by introducing possible differences in their cellular signaling pathways. And, the results of the effect of mTORC1 inhibitors on tumor are rather updated in detail and informative (lines 300 and below), the information on mTORC1 activity and roles in senescence is limited and not well updated. Finally, Figure 1 is not quite explanatory, and needs better legend. These need to be improved.

Author Response

This review summarizes the reports of literature on mTORC1 activity and autophagy in several forms of senescence and examined their functional connections to suggest autophagy regulatory pathways that are different in senescent and normal cells. The authors also tried to check the senescence-specific regulatory link between mTORC1 activity and autophagy in cancer cell context in an attempt to suggest alternative cancer therapeutics.

In general, the manuscript is written well and reasonably organized. However, the literature search appears to be carried out rather poorly and therefore, this review provides limited information and poor insight in the roles of mTORC1 and autophagy and their relationships at molecular level.

R: In order to make our work more comprehensible and insightful, we have extensively edited the manuscript. In particular, a more exhaustive description of the autophagy process is now followed by a description of mechanisms involved in autophagy upregulation in the context of mTORC1 inhibition. Necessarily, we had to touch on the role of lysosome-located mTORC1 in autophagy regulation and cell survival. We hope that these new pieces of information will provide a better account of what is known about mTORC1 and autophagy (and how they are linked at the molecular level) in cells subjected to metabolic stress, before describing what is known about mTORC1 and autophagy in senescent cells. Part of this information was incorporated in Figure 1.

Furthermore, the authors repeatedly mentioned the possible difference in the relationship between mTORC1 and autophagy among different types of senescence but failed in suggesting any clue by introducing possible differences in their cellular signaling pathways.

R: Certainly, there are still gaps in our understanding of the molecular details that underly the functional interaction between mTORC1 activity and autophagy in different models of cellular senescence. However, there are some cautious generalizations we can make, which are now described in the legend of Figure 2:

  1. While normal or primary cells usually contain reduced (although variable) basal levels of mTORC1 activity and autophagy, most cancer cells are characterized by high levels of mTORC1 activity and variable levels of autophagy.
  2. Induction of senescence in normal or primary cells is generally accompanied by an upregulation of both mTORC1 activity and autophagy. Autophagy in this setting may be functionally coupled to mTORC1-dependent synthesis of SASP components, a functional interaction that is required for the implementation of the senescence program. As a consequence, inhibition of mTORC1 or autophagy can block senescence altogether or specifically abrogate the synthesis of SASP components.
  3. Unlike primary cells, cancer cells rendered senescent by pro-senescence chemotherapeutic drugs upregulate autophagy while retaining high levels of mTORC1 activity. In this setting, mTORC1 and autophagic activities may be functionally uncoupled, with each activity contributing independently to the maintenance or survival of senescent cells. As a consequence, inhibition of mTORC1 or autophagy would be expected to reduce survival of senescent cells, exacerbate the senescence phenotype or lead to more subtle changes in SASP profiles, depending on the context.

The results of the effect of mTORC1 inhibitors on tumor are rather updated in detail and informative (lines 300 and below), the information on mTORC1 activity and roles in senescence is limited and not well updated.

R: We are somewhat surprised by the reviewer’s comment. Actually, we do provide a compelling list of references supporting the idea that mTORC1 activity is required for the orchestration of cellular senescence. We point out that senescence can be affected in different ways following inhibition of mTORC1 in primary fibroblasts undergoing oncogene-induced senescence or primary fibroblasts undergoing replicative senescence. This mTORC1-dependent senescence could also be replicated in other models of senescence. We also touched on the idea that high levels of mTORC1 activity could drive cell cycle arrest towards senescence, while low levels of mTORC1 activity could drive cells to other forms of cell cycle arrest, such as quiescence. We also comment on the effects that mTORC1 inhibition has on models of senescence involving mesenchymal stem cells, endothelial cells and cardiomyoblasts. Finally, we comment on the effects that inhibiting mTORC1 has specifically on SASP components.

Finally, Figure 1 is not quite explanatory, and needs better legend. These need to be improved.

R: A full explanatory legend was added to this new version of the manuscript.

Round 2

Reviewer 2 Report

Most of my concern is now appropriately addressed, and the revised is much improved.